



# Production of Definitive Data from Indonesian Geomagnetic Observatories

Relly Margiono[1,3], Christopher W. Turbitt[2], Ciarán D. Beggan[2], and Kathryn A. Whaler[3]

[1]Sekolah Tinggi Meteorologi Klimatologi dan Geofisika, Indonesia
[2]British Geological Survey, Edinburgh, UK
[3]School of GeoSciences, University of Edinburgh, UK

**Correspondence:** Relly Margiono (relly.margiono@stmkg.ac.id)

**Abstract.** Measurement of the geomagnetic field in Indonesia is undertaken by the Meteorology, Climatology and Geophysics Agency (BMKG). Routine activities at each observatory include the determination of declination, inclination and total field using absolute and variation measurements. The oldest observatory is Tangerang (TNG), started in 1964, followed by Tuntungan (TUN) in 1980, Tondano (TND) in 1990, Pelabuhan Ratu (PLR) and Kupang (KPG) in 2000 and Jayapura (JAY) in 2012. One

of the main obligations of a geomagnetic observatory is to produce final measurements, released as definitive data, for each year and make them widely available both for scientific and non-scientific purposes, for example to the World Data Centre of Geomagnetism (WDC-G). Unfortunately, some Indonesian geomagnetic observatories do not share their data to the WDC and often have difficulty in producing definitive data. In addition, some more basic problems still exist such as low quality data due to man-made or instrumental noise, a lack of data processing knowledge, and limited observer training. In this study,

we report on the production of definitive data from Indonesian observatories and some recommendations are provided about how to improve the data quality. These methods and approaches are applicable to other institutes seeking to enhance their data quality and scientific utility for example in main field modelling or space weather monitoring.

## 1 Introduction

A geomagnetic observatory is a permanent site or installation at which the geomagnetic field vector and strength is continuously recorded as a function of time at the Earth's surface. The main objective of an observatory is to record the change of geomagnetic field over both short (e.g. daily variation, magnetic storms) and long (e.g. secular variation) periods. Unfortunately, the spatial distribution of geomagnetic observatories is uneven and they are concentrated in the Northern hemisphere (Rasson et al., 2010). Although recent satellite missions such as CHAMP and Swarm (Friis-Christensen et al., 2006; Reigber

et al., 2002) have extended global coverage of the magnetic field, there is no guarantee that satellites will continue to operate over an extended period. Therefore, installing and maintaining ground observatories is a sensible way to guarantee the continu-



ity of magnetic data records. In addition, satellites measurements are affected by ionospheric current systems and are subject to the time-space ambiguity of satellite measurements.

In order to reach high-quality levels of geomagnetic data useful for scientific purposes it is generally necessary to adhere to standards, for example those defined by INTERMAGNET (St-Louis, 2012) which is an international body whose members

agree to a set of defined rules and metrics for the consistent operation of observatories. Not all geomagnetic observatories are part of this network as there are minimum requirements in terms of the type and quality of observatory instrumentation, data processing and transmission. Currently there are around 130 observatories in this network, but again the spatial distribution is still weighted to the European sector and North American sectors. Observatories which do not reach INTERMAGNET standards can alternatively provide their data to the World Data Centre of Geomagnetism (WDC-G). It is obligatory for each

observatory to send their final definitive data once per year, though they also can provide their data after preliminary processing, which are referred to as quasi-definitive (QD) data (Clarke et al., 2013; Peltier and Chulliat, 2010). In addition, INTERMAG-NET also allows observatories to send their data in near-real-time through Geomagnetic Information Nodes (GINs). Note, these near-real-time data are not definitive either but can be useful in purposes such as space weather applications.

Indonesia has six geomagnetic observatories, which are operated by the Meteorological Climatological and Geophysical

Agency (BMKG). Unfortunately, the data quality is often poor, caused by noise in the measurements and poor data processing techniques. In addition, there is a tendency for slow transmission of the data, so no observatory has yet joined the INTER-MAGNET network. There is also a lack of Indonesian observatory data in the WDC-G which means geomagnetic data in this region cannot be fully utilized. This is particularly unfortunate as the equatorial regions display magnetic field features such as Equatorial Electrojet (EEJ) (Sugiura and Cain, 1966), which are important for geomagnetic index determination (e.g. k-index)

(Mursula and Martini, 2007), and currently exhibit rapid secular variation and acceleration (Kloss and Finlay, 2019). For these reasons, we have made considerable efforts to create definitive data from Indonesian observatories and store them in the WDC-G data portal. This paper aims to report the work undertaken to produce definitive data from Indonesian observatories and provide recommendations for improvement. In addition, we offer it as a template for other institutes wishing to identity issues and improve the quality of data from their observatory network.

## 25  2   Data and Methods

We utilized data from four Indonesian observatories, Tuntungan (TUN), Pelabuhan Ratu (PLR), Tondano (TND), and Kupang (KPG); their locations are given in Table 1. The data consist of raw magnetic measurements from variometers and scalar instruments, as well as absolute observations. For modern geomagnetic observatories, two types of observation are made consisting of absolute measurements and relative measurements or variations. An absolute measurement is manual observation

to determine the true strength and direction of the magnetic field vector in the geographic reference frame at a particular time (typically only once or twice per week) using an absolute magnetometers (e.g. Declination Inclination Magnetometer, Proton Precession Magnetometer), whereas relative measurements are continuous observations (usually once per second) of the





variations in the field vector using an automatically recorded variometer (e.g. Fluxgate magnetometer). This type of instrument is installed on a stable pillar to reduce the effect of movement or drift over time.

Measurements in Indonesia are a combination of several instruments of different type and age depending on the observatory in question. Note due to the large size of the country, the observatories operate as semi-independent services which provide data to BMKG as the central hub and funder. As part of the study we wished (a) to understand the data issues within each of the observatories and (b) to produce a revised set of magnetic field measurements which are as close to definitive data as possible. In order to do this, we requested the raw magnetic data from the observatories by means of an individual direct request. Once we had collected the raw data for 2010 through to 2018, we began to examine each dataset to identify the root cause of any issues and prepare a processing chain to produce definitive data.

**Table 1.** Indonesian observatories which are used to derive definitive data

| Obs | Lat (°) | Lon (°) | Data Years |
| --- | --- | --- | --- |
| TUN | 3.51 | 98.56 | 2016-2017 |
| PLR | -6.98 | 106.55 | 2007-2017 |
| TND | 1.29 | 124.95 | 2009-2017 |
| KPG | -10.20 | 123.67 | 2009-2017 |

## 2.1 Definitive data production

Figure 1 shows a flowchart of the definitive data processing steps we adopted. Producing definitive data begins by collecting absolute and variometer raw data as seen in Table 1. We note each observatory has a different sampling rate depending on the instruments deployed (e.g. one second, five second). We initially tried to produce data in the one-minute IAGA-2002 format[1] by applying the Gaussian filters recommended by INTERMAGNET. We also checked the variometer raw data prior to the conversion to assess whether or not the raw data were noisy (i.e. containing unwanted, artificial signal); any noisy data were removed following the method of Khomutov et al. (2017). Normally, each observatory has a single data logger to capture the variometer and scalar proton data; however, if the loggers are separate, we performed additional processing to ensure both datasets are correctly combined (e.g. aligning by timestamp).

The next stage, baseline calculation, is an important part of the processing, and the stability of the result determines the absolute accuracy of the processed data. Inaccuracy in the processing will affect the final definitive data released. To provide an accurate calculation, first we should understand the orientation of the variometer sensor. The HEZ (Magnetic North, Magnetic East, vertically downwards) orientation is the most popular at a magnetic observatory due to the ease of the installation i.e. it requires just orienting one of the horizontal sensors to the magnetic east direction (indicated by a zero value in the output of the E-sensor) so that the orthogonal sensor is directly pointing to magnetic north. In contrast, some researchers argue that

---

[1]https://www.ngdc.noaa.gov/IAGA/vdat/IAGA2002/iaga2002format.html





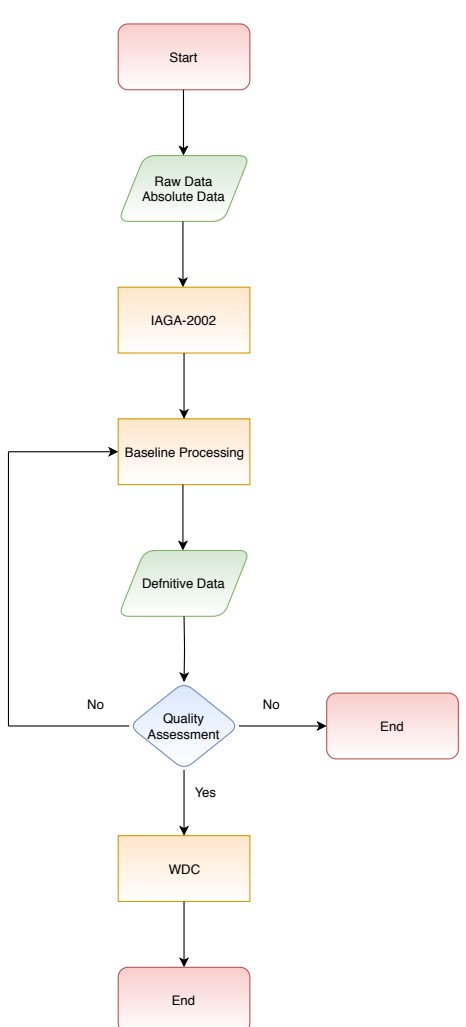

**Figure 1.** Flow chart to produce definitive data and quality assessment





the geographic XYZ (North, East, vertically downwards) orientation will produce a more stable installation because the sensor does not need a readjustment in the future and it is very suitable for observatories that are located at high latitude (Jankowski and Suckdorff, 1996). However, installation in the XYZ direction needs proper initial measurement to determine the geographic north (X) or geographic east (Y) direction prior to the installation and observatories can unintentionally introduce errors during

the installation stage should the sensors be installed with some alignment error (Rasson, 2005). However, Gonsette et al. (2017) show it is possible to determine a baseline from all types of sensor orientation using correction methods even with a non-ideal installation (e.g. an unstable pillar). However, their method needs a high frequency of absolute measurements, which can be made using an automatic observatory (Poncelet et al., 2017) but might be difficult to apply at a normal observatory. In this paper, we will describe how we determined an instantaneous baseline from the HEZ sensor orientation.

The definition of the magnetic field vector and of the magnetometer sensor orientation often leads to confusion because the magnetometer can be oriented in a different position (HEZ, XYZ, DFI) to the field vector. After installation, the magnetometer remains static, but the vector field changes over time. Here, we describe the three orthogonal magnetometer sensors of the fluxgate as P, Q and Z (Figure 2. In this description, both the vertical downward magnetic field vector and vertical component sensor (Z) are the same. Table 2 defines the subscripts used to label the field measurements used in the baseline calculation.

The measurements consist of full-field, part-field, baseline and offset both in the absolute and variometer pillars.

**Table 2.** Definition of field subscripts

| Subscript | Field |
|:---:|:---:|
| A | Full-field at the absolute site |
| S | Full-field at the variometer site |
| V | Part-field as recorded by the variometer |
| B | Baseline |
| D | Site difference between an absolute site and variometer site |
| O | Bias field/sensor offset |

A fluxgate magnetometer only measures a relative variation of the magnetic field. These variation data must be corrected with an offset values in three components, commonly known as the instrument baselines which is include terms accounting for the bias field applied to the sensor, electronic offsets and gradients in the magnetic field vector between the variometer position and the absolute measurement position (Turbitt, 2003). Where a continuously-recording scalar instrument (such as a

proton precession magnetometer or PPM) is also being operated, a similar baseline measurement, known as a site difference, is applied to account for field gradients between the scalar instrument and the absolute measurement position. The site difference can be evaluated by temporarily running a secondary scalar instrument at the absolute measurement position. This value is then used both as a quality reference for the variometer data and in the process of determining the field vector at the absolute measurement position during manual measurements.





Typically, a manual absolute measurement resolves the time-varying field vector in the spherical components set defined as: declination ($D_A(t)$), inclination ($I_A(t)$) and total field ($F_A(t)$). Other commonly used components (e.g. $H_A(t)$ and $Z_A(t)$) can be derived using Equations 1 and 2. The absolute values for $D$ and $I$ are typically computed from the average of 4 measurements made with fluxgate magnetometer attached to a non-magnetic theodolite. Details of how to make the absolute

5    observations and the recommended procedure can be found in Jankowski and Suckdorff (1996); Rasson (2005); Turbitt (2003).

$$Z_A(t) = F_A(t)\sin I_A(t) \tag{1}$$

$$H_A(t) = F_A(t)\sin I_A(t) \tag{2}$$

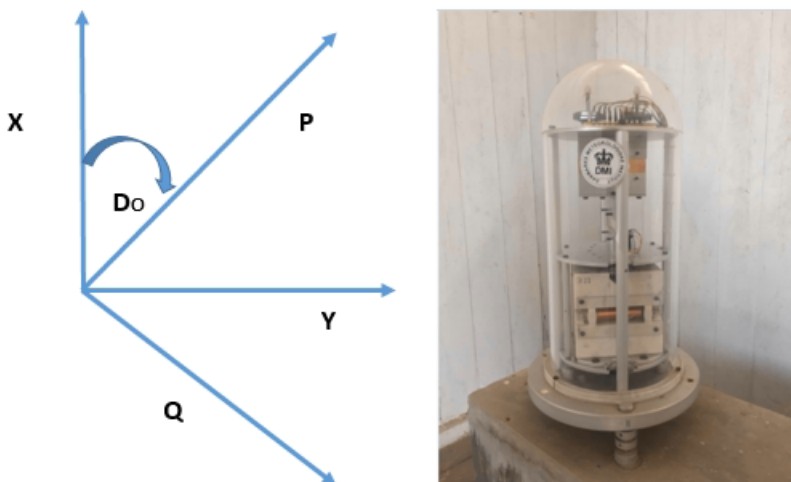

**Figure 2.** Sensor orientation (left) of a typical fluxgate magnetometer (right) in the horizontal plane. P and Q define the orthogonal sensors aligned in the horizontal plane, $D_0$ is the declination, X and Y are geographic North and East.

Detail of baseline calculation for each component (vertical, horizontal and declination components are defined in Appendix **??**). Finally, data quality assessment is performed using IMCDview software[2].

10   **3   Results**

Figure 3 shows definitive data for four Indonesian magnetic observatories (TUN, KPG, TND , PLR). Note that Jayapura (JAY) and Tangerang (TNG) observatories have not been included in the results as their data quality was not good enough to be

---

[2]https://www.intermagnet.org/publication-software/software-eng.php

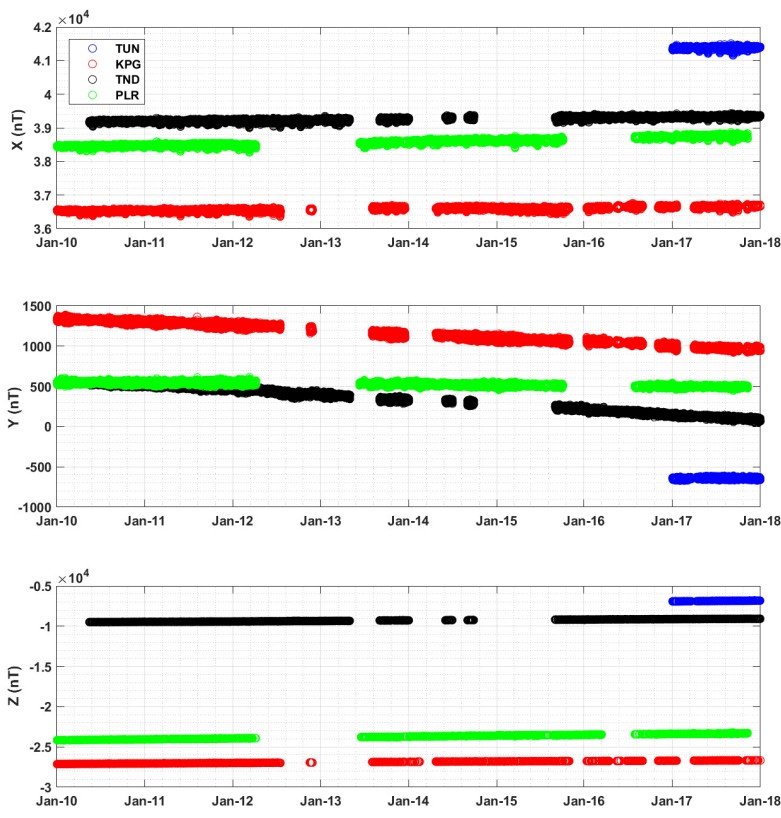

**Figure 3.** Hourly means at Indonesian observatories in the WDC-G data portal for the period 2010-2017

added to the WDC-G. In the case of TNG, the data have problems due to numerous spikes and steps (Figure 4a). This noise is assumed to be associated with electric trains that operate in Tangerang City. The railway, installed in 1997, runs less than 1 km from the observatory. Consequently, the electrical current from the train leaks into the ground and interferes with the geomagnetic data (Pirjola, 2011; Curto et al., 2008). Jankowski and Suckdorff (1996) stated that a geomagnetic observatory

5    should be at least 300 metres from other buildings and 1 km from a railway. If the train system is electric, the distance should be several kilometres (tens of kilometres for DC trains). For this reason, we can conclude that the location of TNG is no longer suitable for recording geomagnetic activity. However, the TNG's night time data are potentially usable as in this period trains are not running so the data are quieter than during the day. JAY is a new observatory and thus is not included due to insufficient data being available at present from this station.

10    Figure 3 shows, generally, from the beginning of 2010, all observatories have excellent data continuity, but from 2012 onward gaps appear in their records. The INDIGO project which helps to build new observatories and supports the existing



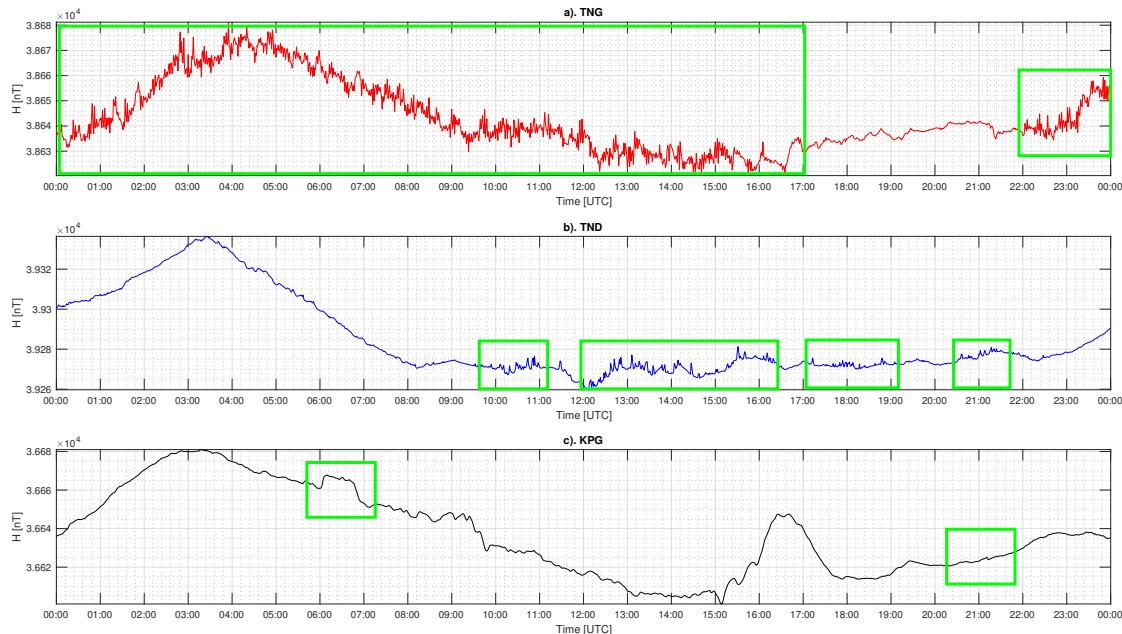

**Figure 4.** Example artificial disturbance (or 'noise') in Indonesian observatory data during 2016. Data are the computed minute-mean values. The green boxes indicate the periods of disturbance.

observatories in Indonesia was launched in early 2010. This project gave digitizers and magnetometers to the Indonesian observatories under coordination of British Geological Survey (BGS) and Institut Royal Météorologique de Belgique (IRM) (Rasson et al., 2010). A couple of years after this project was initiated, problems arose such as sensor and digitizer faults. In addition, the local operators do not have ability to fix the instrument themselves instead returning the instrument to the UK to

be repaired, delaying their return to service. The broken instruments meant the complete observation of magnetic fields could not be carried out and impacted on the data quantity.

Starting in 2016, BMKG initiated a project to standardize the sampling rate of fluxgate data recording. The institution preferred to have a faster sampling rate (e.g. one second) as BMKG wishes to detect magnetic anomalies perhaps associated with events e.g. earthquake precursors (Ahadi et al., 2015). BMKG also required all observatories to use the LEMI format

(Lviv, 2008) as standard so that the old format (INDIGO format) was replaced. The change of data format resulted in a new problem as the LEMI format gives a pseudo-offset in the fluxgate data. This offset produced discontinuities in the baseline record. Consequently, some observatories have had to adjust their baseline calculation to produce the correct baseline. In addition, other problems such as timing inaccuracies between fluxgate sensors and scalar proton magnetometers arose because the new systems were not recording to a single data logger, as was the case with INDIGO equipment. Thus, the observers

needed to manually to fix these issues if the timing discrepancies occurred.



**Figure 5.** Step distribution and step periods at KPG observatory. Panels a,b,c are histograms of the differences in the H,E and Z components respectively, and panel d is the step period. Magenta lines in the difference distributions are the assumed step threshold.

At some of the observatories gaps after 2016 are due to the occurrence of noise in the data. This noise was identified as random noise at TNG (Figure 4a) and TND (Figure 4b) and steps in KPG (Figure 4c). It is difficult to remove the step noise at KPG, as the magnitude of the steps vary every day and there is no fixed pattern so automatic detection via an algorithm is not possible. We constructed histograms of first differences, $diff(i) = (d(i-1) - d(i))$, for a year of minute-mean variations of



KPG data, and assumed that absolute differences > 0.5 nT were most likely steps created by artificial disturbance rather than natural field variations (Figure 5). From these panels, we can see that the difference distribution follows a Laplacian distribution and the majority of differences are within the step threshold. In addition, the step durations in the minute-mean variation data were estimated by eye (Figure 5d) as the steps vary in amplitude (hence why they are difficult to detect automatically). We can

see (Figure 5d) that the duration of a steps is typically less than one hour (the histogram is binned every 60 minutes), from which we can conclude that filtering the data with span of one hour or less will not attenuate the effect of these artificial signals from the time series i.e. the steps will be evident in one-minute and hourly mean values. However, the symmetrical distribution of these disturbances and their relatively short duration mean that we are able filter out these steps by means of daily mean values. We found that the random noise at TND and TNG was difficult to treat, and the best way of dealing with it was to delete

the noisy segments (Khomutov et al., 2017), consequently producing large gaps of missing data.

    To measure how well the Indonesian observatory data monitors secular variation, we compared the data to the main field model of CHAOS-6 model (Finlay et al., 2016) as seen in Figure 6. The secular variation is computed using annual difference of monthly means. From the figure, we can see if the secular variation in vertical component change (in nT/year) having decreasing pattern. TND and KPG secular variation match the overall trend of the CHAOS-6 model, as does KDU as reference.

PLR shows a trend similar to the CHAOS-6 model, but this is dominated by large variations in the secular variation values. This large variation is possibility due to the presence of external magnetic field signal such as Equatorial Electro Jet (EEJ). If this is the case, further processing would be required to isolate secular variation from the PLR data, such as the proxy method (Cox et al., 2018; Brown et al., 2013). Isolating the secualar variation could yield signals such as geomagnetic jerk, for which there is a lack of data in the Indonesian region.

## 4   Improving the entire processing chain


  In Section 2.1, some methods to improve geomagnetic data quality are described. We have applied those methods to assess Indonesian data quality, and found that each observatory has its own specific problems; for each we recommended some improvements that were implemented to enhance the data quality. However, issues related to making the original observations and data processing remain. The major problem we found was in producing definitive data as a final step in the geomagnetic

processing chain. Before this study began, there was a lack of Indonesian definitive data in the WDC portal. Consequently, Indonesian observatory data could not be used for applications such as geomagnetic modelling (e.g. McLean et al., 2004; Gillet et al., 2015) although recent satellite missions can help partially overcome this problem.

    Absolute geomagnetic observations at the magnetic observatory require proper care and attention. For absolute observations, the role of the observer is a vital consideration in order to produce high data quality. An observer needs suitable knowledge

about the technical aspects of absolute measurement and an adequate understanding of the science of the magnetic field is also necessary. We found that some observers at Indonesian observatories only focused on the technical aspects, regardless of the fundamental science, causing mistakes to be made. For example, some of them still had magnetic materials on them while performing absolute observations (e.g. mobile phones or jewellery); in fact, these materials clearly contributed to measurements

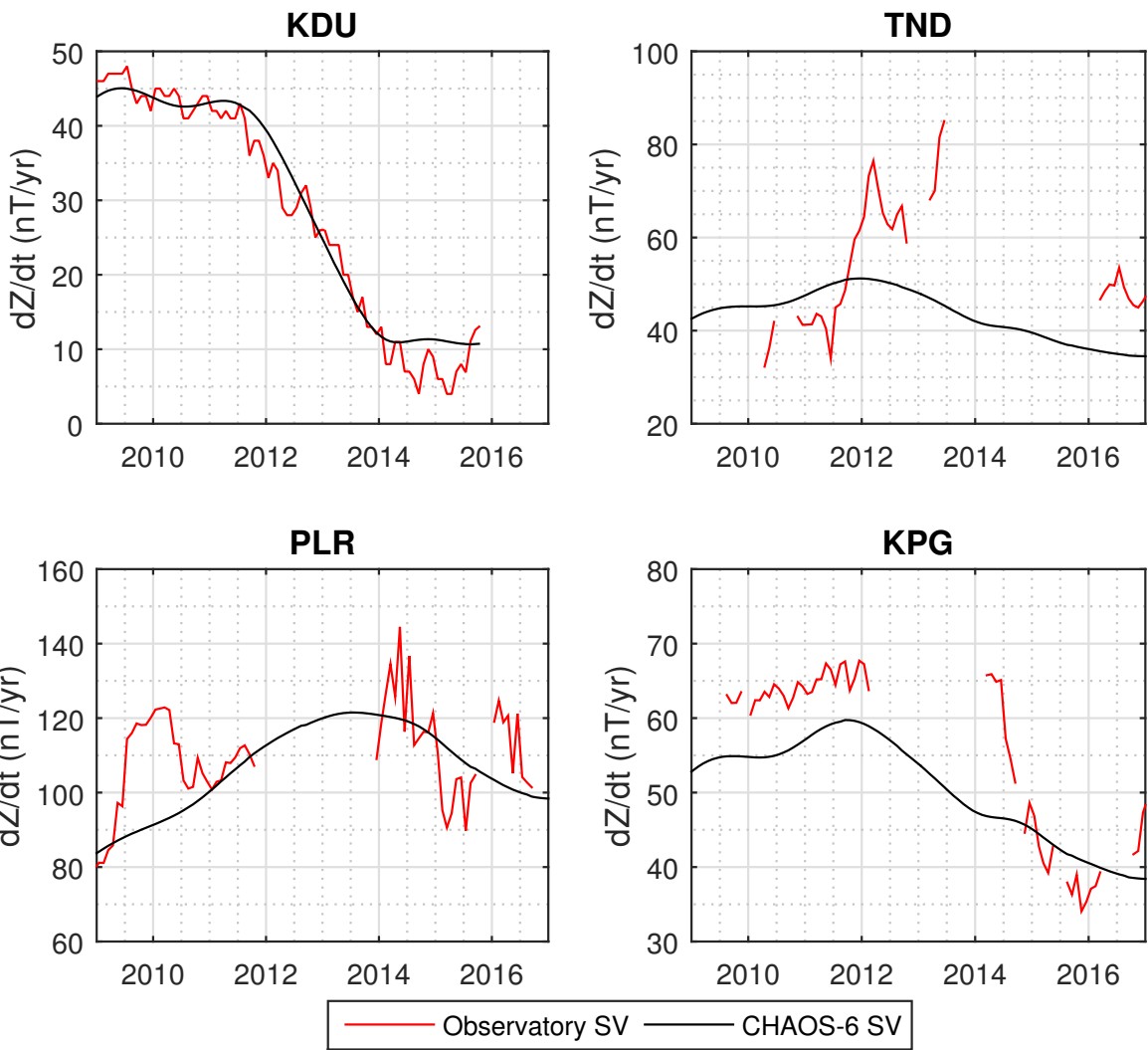

**Figure 6.** Secular variation comparison between Indonesian Observatory Data (KPG, PLR and TND) and Chaos-6 model (Finlay et al., 2016). Kakadu (KDU) is presented as reference observatory near to Indonesia

errors. The only way to make better absolute measurements is by considering all aspects both technical and practical and improving the operator skill level; subsequently a series of workshops and seminars were organised to educate and inform the observers and encourage them to raise their own standards.

Variation measurements are the other form of observation at an observatory. These observations mainly depend on the performance of the automated instruments which measure the magnetic field continuously. The basic requirement to achieve excellent recording is to locate the sensor at a quiet place so that it only measures the natural magnetic field. Some Indonesian





geomagnetic observatories are not located in ideal places, and this problem will only be overcome by moving the observatory to a better location with initial and proper site surveying prior to installation. In addition, the sensor orientation and related installation techniques have to be considered. A poor installation can have an impact on baseline processing and ultimate on the absolute accuracy of the published definitive data. The HEZ orientation is popular in our observatories as this is easy to install

and, as shown, the derivation of instrument baselines is relatively straight-forward. However, one observatory does use the XYZ orientation which is more challenging to process as any error in installation is difficult to compensate for in post-processing.

Geomagnetic data processing is the next stage after absolute and variation observations are performed. The primary sources for this processing are raw data from absolute and variation observation. These raw data are then processed to produce final definitive data. We found that Indonesian observatories did not use standard processing tools. At some observatories, data

processing relied on manual processing assisted by use of a computer spreadsheet, while others had established their own data processing using tools written in common programming languages. The lack of a data processing standard made the final definitive data production difficult, especially for the observatories that still use manual calculation. Although they produced daily means, monthly means and annual means, the data format was not standardised. As manual processing is slow, it cannot handle more challenging tasks such as data conversion, data merging and transmission. To remedy this, we introduced a user-

friendly software, namely GDASview [3], developed by the British Geological Survey, to help automate data processing at Indonesian observatories.

Finally, the data from four Indonesian observatories are now available at the WDC data portal. These definitive data still have limitations such as data gaps and less accurate baselines, but the quality is reasonable enough to be made available to the international community. The justification of this is based on visual analysis, data quality checks, and the implementation of

a standardized method to produce definitive data. We hope the data quality will improve and all the Indonesian observatories will obtain INTERMAGNET certification in the near future. In addition, robust data quality measurements which include more numerical calculation are recommended, such as using momentary values (Curto and Marsal, 2007) or a more statistical approach (Zhang et al., 2016), but both methods rely on complete data series. This should be accomplished in the next few years. We expect that this work has impact for the science of geomagnetism e.g. the data can be used to derive the next IGRF

model, or to observe detailed secular variation and acceleration around the equatorial region, such as in the study by Kloss and Finlay (2019). There are fewer observatories in the southern hemisphere and over the Pacific, so availability of quality Indonesian observatory data would fill a significant gap. In addition, the data availability would improve analysis of jerks in the western Pacific region (Torta et al., 2015). Furthermore, this work also can be used as a reference for Indonesian observatories or other magnetic observatories which have the same problems to improve their data quality (e.g. better observations and data

processing). We have produced some recommendations for Indonesian Geomagnetic observatories below.

---

[3]http://www.geomag.bgs.ac.uk/code/GdasView/launch.html





## 5  Conclusions

In this paper, a new set of definitive data production for four Indonesian observatories is described and presented. We have produced definitive data from the observatories using the methods described to create a standardised high quality set of measurements for scientific exploitation. We explain the steps taken to improve upon the data collections and processing protocols

that previously existed for the four main observatories in the network. From this we identified issues in the manner in which absolute and relative measurements were being made and suggested improvements in processing and training to enhance the quality of the magnetic time-series. The final definitive data have been published at World Data Centre for Geomagnetism (Edinburgh) [4]. The data can fill the gap in the Pacific region and provide input into geomagnetic modelling and secular variation studies. In the next few years, Indonesian geomagnetic observatories should maintain and enhance their quality, with the main

institution BMKG taking responsibility for ensuring continued improvement.

*Data availability.*  The final definitive data have been published at WDC (http://www.wdc.bgs.ac.uk/dataportal/)

## Appendix A:  Baseline Calculation

### A0.1   Vertical Component Baseline

The vertical sensor (Z), which is aligned vertically downward, measures the variation of the magnetic field at the variometer

site ($Z_S(t)$). Upon installation ($t_0$), a bias field is applied to this sensor with the same magnitude but opposite direction to that of the main field, so the initial value is close to zero. At any time of measurement, the field is thus defined as $Z_V(t)$. The P and Q sensors are located in the horizontal plane, and the P sensor makes an angle $D_0$ with true north (X) as shown in Figure 2. In the installation, there is no bias applied to the orthogonal Q sensor because this sensor is rotated to the position where its value is zero (i.e. indicating the Q sensor is oriented in the magnetic east direction). This procedure will align the P sensor

to the instantaneous horizontal field (magnetic north) at the variometer site $H_S(t_0)$, and the angle, $D_0$, will approximate the instantaneous value of declination $D_S(t_0)$. In the initial installation, the bias field is applied to the P sensor, so the output of the sensor in any point measurement is $P_V(t)$.

The Z baseline value can be determined by subtracting the absolute Z value ($Z_A(t)$) from the variation $Z_V(t)$. In this calculation, the Z baseline also can be determined as the sum of the sensor offset ($Z_O(t)$) and the site difference $Z_S(t)$.

$$Z_B(t) = Z_A(t) - Z_V(t) \tag{A1}$$

---

[4]http://www.wdc.bgs.ac.uk/dataportal/





## A0.2   Horizontal Component Baseline

The H baseline can be processed similarly to the Z baseline, but because both sensors (P and Q) cannot be assumed aligned or orthogonal to the H vector after the initial installation (owing to the field changing with time), the processing is slightly different. $H_S(t)$ can be determined from P and Q sensor values using vector relationships as given by Equation A2.

$$H_S^2(t) = P_S^2(t) + Q_S^2(t) \qquad (A2)$$

The equation then can be re-expressed in terms of variometer output,

$$H_S^2(t) = (P_V(t) + P_O(t))^2 + (Q_V(t) + Q_O(t))^2. \qquad (A3)$$

As noted before, there is no offset applied to the Q sensor, so equation (A3) becomes

$$H_S^2(t) = (P_V(t) + P_O(t))^2 + (Q_V(t))^2. \qquad (A4)$$

Equation (A4) then can be written in terms of the field at the absolute site by adding the site difference.

$$H_A^2(t) = (P_V(t) + P_O(t) + P_D(t))^2 + (Q_V(t) + Q_D(t))^2 \qquad (A5)$$

The site difference of the Q sensor can be approximated to zero assuming the observatory is placed at a magnetic clean site with low gradient. In addition, $P_V(t) + P_O(t)$ can be combined in terms of the baseline ($P_B(t)$),

$$H_A^2(t) = (P_V(t) + P_B(t))^2 + Q_V(t)^2. \qquad (A6)$$

Finally, the baseline can be defined as,

$$P_B(t) = \sqrt{(H_A(t)^2 - Q_V(t)^2)} - P_V(t) \qquad (A7)$$

In most conditions, the value of $Q_V(t)$ is small so that $P_B(t)$ can be approximated as

$$P_B(t) = H_A(t) - P_V(t). \qquad (A8)$$





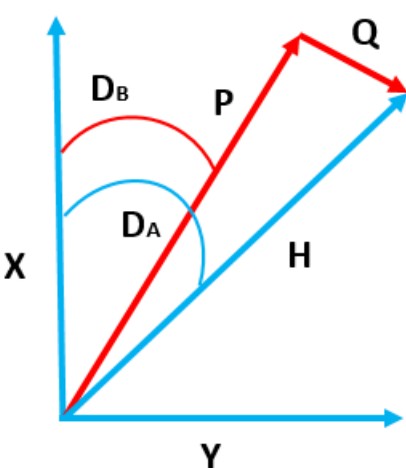

**Figure A1.** Variation of declination angle at the variometer sensor. The difference between $D_A$ (Absolute declination) and $D_B$ (Baseline declination) is the declination variation in the variometer sensor.

### A0.3 Declination Component Baseline

The declination baseline, $D_B(t)$, is an imaginary angle at the absolute site between absolute declination $D_A(t)$ and variation (angle between $Q_A(t)$ and $H_A(t)$) as shown in Figure A1. If the variometer is aligned such that the P sensor makes an angle ($D_B$) to the true north, the angle of declination at the absolute site can be determined as

$$D_A(t) = D_B(t) + \sin^{-1}(Q_A(t)/H_A(t)). \tag{A9}$$

This equation can be written in terms of the variometer site:

$$D_A(t) = D_B(t) + \sin^{-1}[(Q_V(t) + Q_O(t) + Q_D(t))/H_A(t)] \tag{A10}$$

Because the values of $Q_O(t)$ and $Q_D(t)$ are very small (and unknown), we can make the approximation

$$D_A(t) = D_B(t) + \sin^{-1}[(Q_V(t)/H_A(t)] \tag{A11}$$

and the final baseline calculation can be written

$$D_B(t) = D_A(t) - \sin^{-1}[(Q_V(t)/H_A(t)]. \tag{A12}$$





The exact time of baseline calculation depends on the four circle readings of absolute measurement made using the theodolite, so the variometer data captured at the same time should also be averaged. If the baseline does not vary significantly, the baseline equation can be derived from the variometer data as in Equation (A12).

**Appendix B:  Recommendations for Indonesian Geomagnetic Observatories**

BMKG is the central institution for all geomagnetic observatories in Indonesia. If it can make good policy or provide standard operational procedures, it will have a strong impact on the quality of magnetic observatories in remote areas. Table B1 shows the summary recommendations from the central institution.

**Table B1.** Summary recommendations for Indonesian observatories

| Recommendation |
| --- |
| Data Collection |
| Common Data Format |
| Common Data Quality Checking |
| Metadata |
| Common Data Processing Platform |
| Training and Intercomparisson |
| Common Data Publishing (Network) |

First, it is advisable to centralise the collection of data. Currently, this stage is achieved by the collection of raw data using file transfer protocol (FTP) from the magnetic observatory to the central BMKG office. This data collection can be extended

by including absolute observation data so that the central institution can make comparison results for the baselines.

Second, it is advisable to introduce standardization of data reporting, including common data formats and sampling rates. This issue has been achieved by standardisation of all magnetic observatory raw data to one-second data using the LEMI format. These raw data should be converted to the standard magnetic data format (e.g. IAGA-2002 format) after the raw data are processed and definitive data are produced.

Third, centralised quality control of data should be implemented. This can be done by day-to-day checking of the quality of raw data and weekly checking of absolute measurements. If the central institution finds an error in the raw data (noise, jumps, missing data, abnormal data), it should be confirmed with the observatory to determine what happened so that a quick solution can be found. In addition, BMKG also has to confer with the observatory if the absolute observation results are less accurate or a jump is identified in the baseline. It can request an alternative/repeat of the absolute measurement. This aims to encourage

observers to make absolute observations carefully. BMKG also should request additional measurements from the observatory,



such as a site difference measurement, as this measurement is very important. A three month interval is appropriate and the results should be documented in the metadata.

Fourth, complete metadata should be stored at the central institution. The metadata consist of observatory instrument records, diaries of observatory changes, latitude and longitude, contact person, address, updated instrumentation, bulletin, site plan.
These metadata should be updated as soon as there are any changes at the observatory. These metadata are very important, especially if there is a problem in the data processing, as inaccurate metadata can produce errors in the data processing (e.g. sensor orientation).

Fifth, BMKG should provide a common data processing procedure. This procedure should state how to process raw data, absolute data, baseline calculation and data transmission. These procedures can be accomplished by providing user-friendly soft-
ware or a calculation template. BMKG should provide a guide and tools for standard data processing software (e.g. GDASview, MAGPY) or MATLAB scripts to be implemented at magnetic observatories. All the observatories should have a similar data processing procedure so that the output of the processing is standard. The implementation of the data processing can be given by on-site training or in a workshop.

Sixth, training of the observers should be take place at least once a year. This training should attract observer representation
from all Indonesian observatories. In the training or workshop, it is necessary to make an inter-comparison of instruments, simultaneous absolute observations and practise data processing. The aims of this training are to produce standard approaches for all the observers, and to build a network of colleagues that can advise and help each other.

Lastly, the observatory should publish their definitive data to the WDC so that the data can be used for international community. In addition, each observatory should aspire to join INTERMAGNET or become certified so that its data quality will be
assessed continuously by the organization.

*Author contributions.* RM initiated this paper publication which is related to his MPhil studies. CWT guided the definitive data processing, including providing training in relevant software use, and CDB and KAW helped in discussion.

*Competing interests.* The authors declare no competing interests.

*Acknowledgements.* We thank all Indonesian observatory staff who helped with data requests and discussions (TUN: Yosi Setiawan, TND:
Edward Mengko, M. Zulkifli, Imam Muslih, KPG: Netrin, Ricky Daniel, Aditya, PLR: Andi, Sofyan, Arafah, TNG: Dinda, Fani, JAY: Jambari, Central BMKG: Mahmud Yusuf, Aziz, Shirodjudin, Suaidi Ahadi. RM also thanks Mr. John Riddick who provided advice and useful discussions during this study. The research reported in this paper was undertaken produced during the M.Phil program of RM at the University of Edinburgh under grant contract PRJ-306 /LPDP.3/2017 from Indonesian Endowment Fund for Education (LPDP), Ministry Finance of Indonesia.



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
