# Peer review of "Production of Definitive Data from Indonesian Geomagnetic Observatories"

_Geoscientific Instrumentation, Methods and Data Systems, 2021_

## Author Comment (AC2)

Response to Reviewer #1:

Firstly, we want to say thank you to Reviewer #1 for their positive comments and suggestions. Below are our responses.

*The manuscript 'Production of Definitive Data from Indonesian Geomagnetic Observatories' by Relly Margiono, Christopher W. Turbitt, Ciarán D. Beggan and Kathryn A. Whaler deals with the production of definitive data from Indonesian geomagnetic observatories.*

*The manuscript is an important contribution and absolutely worthy of publication. I have some comments that I believe will help improve the manuscript (see below). There is, however, one point that I would like to have addressed before I can recommend publication: How stable is the temperature and the F difference for the individual variometers? This will be important to understand the data quality and the data's suitability for Sq studies. I suggest to add a plot that shows, from 2010.0 to 2018.0, the daily range of temperature variation (as well as the monthly range of temperature variation) of fluxgate sensor and electronics of the variometer that was used. Also add a plot of daily mean F differences (i.e. F from calibrated variometer data minus F from scalar magnetometer) and show also the daily range of F differences.*

Response: Many thanks for your question and suggestion. Previously, we did not analyse the temperature as the IAGA-2002 format data does not provide information about the temperature both for the fluxgate and electronics. We agree a temperature analysis would help understand the data quality as the fluxgates are temperature-sensitive. For the updated version for our paper, we have added in a temperature analysis using the raw outputs of the fluxgate data as seen in Figure 5 to Figure 8 in this response (below). In the raw data (INDIGO format), there are three temperature categories: the internal temperature of the INDIGO digitizer box, the temperature of the DMI fluxgate electronics box and the temperature of the DMI fluxgate sensor head. Unfortunately, only the INDIGO digitizer temperature is recorded in our data. As the sensor, the digitizer and the electronic are placed in the same building, the digitizer temperature will be fairly representative of the actual fluxgate temperature, and should indicate its variability.

It is obvious that the temperature data prior to 2017 shows a wide range daily and monthly temperature especially for PLR and KPG. This means that the variometer buildings were not set up properly and the buildings were not temperature insulated. In the mid of 2016 for PLR, and early 2017 for KPG, the daily and monthly temperature range were better. In addition, the temperature range for TND remains steady from 2010 to 2017. In the recommendation section, we add a new recommendation for BMKG to maintain the stability of temperature in the variometer buildings because it can affect the data quality.

Regarding the F difference for the individual variometers, we have added in the plots in the updated version. The plots can be seen in Figure 1 to Figure 4 (below). The daily mean of F differences are close to zero and the daily range of F differences are below 2 nT. It means that the data quality is good enough.

*The abstract states that TNG started 1964. In the WDC, there is hourly data from 1957 to 1964. Why already since 1957? Please also check other observatories for consistency.*

Thank you for you correction. Yes, after looking for details in the history of Indonesian geomagnetic observatories, we found that Tangerang (TNG) started to record the

magnetic field in 1957. We now refer to the observatory website (http://stageof.tangerang.bmkg.go.id/?page_id=15). However, no data have been provided to the WDC since 1969. This has been clarified in the text.

*Please state in the abstract that data from four observatories dating from 2010? to 2018 has been processed and are now published in the WDC*

We have added it in the updated version.

*p 1*

*l. (line) 5: measurements -> versions of the data*

We have changed the sentence in the updated version.

*p 2*

*l. 8 to 10: please move the sentence on WDC down to line 13/14, otherwise the reader might think that WDC accepts QD. Submitting data to WDC requires absolute measurements and calibration of the data, please state this explicitly.*

We have changed the sentence in the updated version.

*l. 17: WDC-G, which*

We have changed the sentence in the updated version.

*l. 19: do you see a link between the EEJ and K indices? I would remove the information regarding K indices.*

We have changed the sentence in the updated version.

*l. 29: is a manual*

We have changed the sentence in the updated version.

*p 3*

*l. 8: How does 2010 relate to the numbers in Table 1 (2016, 2007, 2009, 2009*

The data used varies in year due to the availability of both variation and absolute data. If the data only have variations without absolute values, we did not process the data. We have changed the sentence in the updated version to make it clear.

*Table 1: since you discuss EEJ, please add geomagnetic latitude*

We have added the geomagnetic latitude in the updated version.

*p 5*

*l. 2: which formula were used to calibrate the HEZ orientation? (Linear formula require more frequent realignment of the sensor than more accurate, non-linear formula.)*

The formula to calibrate the HEZ orientation is described in the Appendix A.

*l. 17: baseline, which include*

We have changed the sentence in the updated version.

*note: HEZ baselines also account for the rotation of the variometer*

*Move Table 2 to Appendix?*

We have moved the table into the appendix.

*p 6*

*l. 4: with a fluxgate*

We have changed the sentence in the updated version.

*l. 9: Appendix ??*

We have changed the sentence in the updated version.

*Figure 4a: please include date*

We have added the date in the updated version.

*Figure 5, figure caption: Would be nice to have better explanation:*

*What is step distribution? What is step period? 'differences' should read 'first differences'*

We have changed the paragraph to clarify the meaning. State distribution should be rate of change distribution or first differences distribution. Step period is the period of step in the minute-mean data.

*p 10*

*l. 16: equatorial electrojet (EEJ)*

*Typically, yearly differences of monthly means are very good in suppressing EEJ signals.*

*Reading your text, I assumed that PLR is an EEJ observatory, but it turns out that TUN and TND are closer to the magnetic equator. Could you add a map of the stations and geomagnetic latitudes, preferably QD latitudes? Also, I am not sure why you state that PLR fits the CHAOS SV signal worse than the other observatories, comparison between observatories would be easier if the panels in Figure 6 have the same scale. State in figure caption that you plots annual differences of monthly means.*

We have added a map of Indonesian Geomagnetic Observatories in the updated version and we used World Magnetic Model (WMM) 2015 to display the magnetic equator. In the updated version, we also add root mean square error (RMSE) to support the statement about which observatory has a better fit to the CHAOS-6 model. In the figure caption, we add a sentence that the SV is computed using annual difference of monthly means.

*l. 23: What does 'original' refer to, not sure what this sentence means.*

We have changed the word 'original' to absolute observation.

*Appendix A:*

*p 14*

*l. 14: Do you combine Pv and Po in the term Pb, or do you just rename Po to Pb? Looks like you did the latter.*

Pb should be represent the addition between Po and Pd. We have changed this in the updated version.

*l. 17: You linearise (A7) to get (A8). Is that useful? If you use this linear version and you have strong secular variation of decliantion, then you might have to rotate the variometer every few years to keep Qv small and thus introduce unnecessary jumps in the variometer datat*

You are right that the linearised form (equation A8) is not necessary and introduces errors when the Qv is large (with secular variation or during a magnetic storm for example). Our data processing routines use equation A7, so equation A8 has been removed in the updated version.

*Appendix B:*

*p 17*

*l. 14: should take place*

We have changed the sentence in the updated version.

*References:*

*The Reigber et al., reference has a doi: https://doi.org/10.1016/S0273-1177(02)00276-4*

*I assume the Pocelet et al. reference also has one.*

*Fix the dot in Rasson et al.*

*Fix tech -nique in Rasson.*

*Please check all your references, complete the bibliographic information and bring it in shape.*

We have modified those references with doi and url in the updated version.

**I.  UPDATED FIGURES**

[Figure]

Figure 1. Daily mean F differences and daily range of F differences of KPG observatory.

[Figure]

Figure 2. Daily mean F differences and daily range of F differences of PLR observatory.

[Figure]

Figure 3. Daily mean F differences and daily range of F differences of TND observatory.

[Figure]

Figure 4. Daily mean F differences and daily range of F differences of TUN observatory.

[Figure]

Figure 5. Daily and Monthly Temperature Range of PLR observatory.

[Figure]

Figure 6. Daily and Monthly Temperature Range of TND observatory.

[Figure]

Figure 7. Daily and Monthly Temperature Range of TND observatory.

[Figure]

Figure 8. Daily and Monthly Temperature Range of TND observatory.